# Molecular Effects of Li^+^-Coordinating Binders and Negatively Charged Binders on the Li^+^ Local Mobility near the Electrolyte/LiFePO_4_ Cathode Interface within Lithium-Ion Batteries

**DOI:** 10.3390/polym16030319

**Published:** 2024-01-24

**Authors:** Po-Yuan Wang, Tzu-Heng Chiu, Chi-cheng Chiu

**Affiliations:** 1Program on Smart and Sustainable Manufacturing, Academy of Innovative Semiconductor and Sustainable Manufacturing, National Cheng Kung University, Tainan 701, Taiwan; m58121013@gs.ncku.edu.tw; 2Department of Chemical Engineering, National Cheng Kung University, Tainan 701, Taiwan; ms0220429@gmail.com; 3Hierarchical Green-Energy Materials (Hi-GEM) Research Center, National Cheng Kung University, Tainan 701, Taiwan; 4Fire Protection and Safety Research Center, National Cheng Kung University, Tainan 711, Taiwan

**Keywords:** lithium-ion battery, electrolyte–cathode interface, functional binders, molecular dynamics

## Abstract

The development of lithium-ion batteries (LIBs) is important in the realm of energy storage. Understanding the intricate effects of binders on the Li^+^ transport at the cathode/electrolyte interface in LIBs remains a challenge. This study utilized molecular dynamics simulations to compare the molecular effects of conventional polyvinylidene difluoride (PVDF), Li^+^-coordinating polyethylene oxide (PEO), and negatively charged polystyrene sulfonate (PSS) binders on local Li^+^ mobility at the electrolyte/LiFePO_4_ (LFP) cathode interface. By examining concentration profiles of Li^+^, three different polymer binders, and anions near Li^+^-rich LFP and Li^+^-depleted FePO_4_ (FP) surfaces, we found a superior performance of the negatively charged PSS on enhancing Li^+^ distribution near the Li^+^-depleted FP surface. The radial distribution function and coordination number analyses revealed the potent interactions of PEO and PSS with Li^+^ disrupting Li^+^ coordination with electrolyte solvents. Our simulations also revealed the effects of non-uniform binder dispersions on the Li^+^ local mobility near the cathode surface. The combined results provide a comparative insight into Li^+^ transport at the electrolyte/cathode interface influenced by distinct binder chemistries, offering a profound understanding of the binder designs for high-performance LIBs.

## 1. Introduction

Lithium-ion batteries (LIBs), owing to their high power and energy densities, have emerged as indispensable energy storage devices for various applications, ranging from portable electronic devices to electric vehicles and smart grid systems [1,2,3,4]. The efficacy of LIBs hinges on various components, where binders play a critical role in ensuring the structural integrity and long-term functionality of the electrodes [5,6,7,8,9,10]. Despite their low weight ratio (≤10 wt %) in the electrode composition, the binders help sustain the electrochemical processes within the LIBs [8]. The main function of binders is to act as molecular glue, firmly anchoring active materials and conductive agents to the current collector [10]. This binding function is critical in maintaining the electrode’s structural cohesion during the dynamic charging and discharging cycles of LIBs.

Polyvinylidene difluoride (PVDF) has been a conventional binder widely employed for its electrochemical stability and adhesion properties [11,12]. However, challenges associated with PVDF, such as the use of N-methyl-2-pyrrolidone (NMP) solvent in the slurry fabrication process, have spurred the effort toward developing alternative binders that are environmentally friendly and compatible with aqueous-based fabrication processes [7]. Due to its toxicity and environmental impact, the usage of NMP in PVDF slurry preparation raises concerns about the health risks during the fabrication process. The shift toward aqueous-based fabrication processes eliminates the need for harmful organic solvents, aligning with the growing emphasis on sustainable and health-conscious practices in materials’ synthesis for electrochemical applications [7,8]. Carboxymethyl cellulose (CMC), styrene-butadiene rubber (SBR), and polyacrylic acid (PAA) are emerging as noteworthy examples of such aqueous binders, offering potential solutions to the environmental and health challenges associated with PVDF slurry fabrication [9,10,13,14,15,16].

Recent developments of innovative binder systems have also focused on materials that can both provide strong adhesion and facilitate efficient ion transport, thereby addressing limitations posed by traditional binders like PVDF [5,6,7,8]. Ion-conducting polymers with ethylene glycol moieties, sulfonate groups, carboxylic groups, or nitrile groups have shown promising potential as effective binders for LIBs [7,8,17,18,19]. Notably, polyethylene glycol and sulfonate groups have emerged as highly effective ion-conducting segments [17,18,19,20,21,22,23]. Sulfonated polymers, e.g., Nafion and sulfonated-polyether ether ketone (PEEK), have been shown to effectively reduce polarization near a cathode, thereby improving overall battery efficiency [19,22,23]. The sulfonate groups can serve as base units for single-ion conductors, aiding Li^+^ transport via electrostatic interactions. Binders containing ethylene glycol groups have also been shown to minimize the interaction between Li^+^ and anions and to facilitate Li^+^ transport through high polymer chain mobility. The use of Li^+^-conducting polymers, including polyethylene oxide (PEO), polyacrylonitrile (PAN), and poly(N-vinylformamide) (PNVF), as binders demonstrates reduced charge transfer resistance and enhanced capacity retention in LIBs, signaling the promising development of advanced binder materials for high-performance lithium-ion batteries [17,18,24,25,26].

In our prior studies, we combined experimental and computational approaches to investigate the effectiveness of ethylene oxide (EO) and sulfonate groups in reducing interfacial impedance at the LiFePO_4_ (LFP) cathode [20,21]. These studies illuminated the superior efficacy of sulfonate groups over EO in enhancing battery performance, with sulfonate binders exhibiting greater effectiveness in reducing the concentration polarization near the interface [20,21]. However, the pivotal question remains largely unexplored, such as the correlation between the molecular properties of binders with a minimal amount and the overall impedance at LIB cathodes. In this work, we employed molecular dynamics (MD) simulations to examine the effects of binders at the electrolyte/cathode interface, focusing specifically on the LFP cathode. We selected polyethylene oxide (PEO) and polystyrene sulfonate (PSS) as representative binders due to their relevance to two common functional binder designs: Li^+^-coordinating binders and negatively charged binders, respectively [20]. PEO is known for its ability to coordinate with lithium ions and form stable complexes, whereas PSS interacts with lithium ions via its negatively charged sulfonate groups. By comparing the behavior of these two representative polymers in both liquid electrolyte and solid polymer electrolyte systems, we aimed to unravel the complex effects of binders on Li^+^ transport near the cathode surface. Through an analysis of Li^+^ distributions, Li^+^ coordination, and local Li^+^ diffusivity, our results provide the molecular mechanisms of the intricate binder influences at low loading levels, providing crucial insights for optimizing LIB cathode design and performance.

## 2. Methods

### 2.1. Electrolyte/Cathode Systems

In this study, we employed molecular dynamics (MD) simulations to probe the influence of various binders and electrolytes on the Li^+^ local transport properties near the electrolyte/cathode interface. We specifically focused on the structural and dynamical aspects within both liquid electrolyte (LE) and solid polymer electrolytes (SPE), respectively. The molecular structures for all the molecules, ions, and cathode materials are shown in Figure 1. The LE systems consisted of 1 M lithium hexafluorophosphate (LiPF_6_) solvated in a 1:1 volume ratio mixture of ethylene carbonate (EC) and diethyl carbonate (DEC), a typical LE composition for a lithium-ion battery. For the SPE system, 48 polyethylene oxide (PEO) chains, each with 64 monomers, were mixed with lithium bis(trifluoromethanesulfonyl)imide (LiTFSI) salt in the oxygen/lithium ratio (O/Li) of 16, which is the common experimental formulation for LIB applications.

The cathode material, LiFePO_4_ (LFP), was modeled using a crystalline comprising 3 × 8 × 6 unit cells with the [010] surface normal to the z-axis. The transport of lithium ions along the [010] channel is known to have the lowest diffusive barrier for Li^+^ transport [27]. Additionally, to model the Li^+^-depleted cathode surface, a single layer of FePO_4_ (FP) crystalline cells was introduced on the surface of the LFP cathode by removing Li^+^ with balanced charges of Fe as illustrated in Figure 1.

Three types of polymers, namely polyvinylidene fluoride (PVDF), polyethylene oxide (PEO), and polystyrene sulfonate (PSS), were introduced at the electrolyte/cathode interface for both LE and SPE systems. The polymers applied on each surface of the cathode for different binder systems were (1) 3 PVDF chains, each with 16 monomers, (2) 4 PEO chains, each with 16 monomer units, and (3) 2 PSS chains, each with 10 monomers. The amount of binders corresponded to the weight percent of 5–10 %, a commonly used experimental formulation [20]. The similar polymer loading also allowed us to directly compare the effectiveness of different binders on the Li^+^ mobility near the cathode surface. Note that in the SPE with PEO binder system, since PEO was essentially the polymer hosts of SPE, no extra PEO polymers were added at the electrolyte/cathode interface.

### 2.2. Simulation Details

All components in the system, including the polymers, organic solvents, lithium salts, and LFP, were described with the OPLS (all-atom optimized potentials for liquid simulations) force field with revised partial charges, OPLS^R^ [28,29,30,31,32,33]. The LFP atomic charges were revised based on the molecular model by Smith et al. [27]. The initial configuration for each system consisting of an LFP cathode and liquid electrolyte or solid polymer electrolyte was constructed using the PACKMOL software [34]. The simulation was carried out by Large-scale Atomic/Molecular Massively Parallel Simulator (LAMMPS) software with an integration time step of 1 fs [35]. Periodic boundary conditions were applied in all dimensions. The van der Waals and short-range electrostatic interactions were calculated with a 1.2 nm cutoff. The long-range electrostatic forces were evaluated using the Ewald method [36]. The system temperature was maintained with the Nose–Hoover thermostat [37,38,39]. During the simulation, the FePO_4_ lattice atoms were fixed to maintain the structural integrity of the cathode. Each system was first energy minimized using the steepest descent algorithm. The systems were subsequently pre-equilibrated at 898 K for 100 ps under the NVT ensemble and then annealed to 363 K with an annealing rate of 0.0535 K/ps. A 200 ns NVT simulation at 363 K was then conducted to relax the system. Finally, an 80 ns production run was conducted under the NVT ensemble.

## 3. Results and Discussions

### 3.1. Li^+^ Distribution and Diffusivity near Liquid Electrolyte/Cathode Interface

The concentration profiles of Li^+^, polymer binders, and ${\mathrm{PF}}_{6}^{-}$ for the LE/cathode system are depicted in Figure 2. Based on the concentration profiles, we defined the interface region as the region between the LFP or FP surface and the first trough in the Li^+^ concentration profile. The resulting region thicknesses were approximately 7 Å to 12 Å. For the Li^+^-rich LFP surface, the equilibrium concentration profiles suggest that all the tested binders, including PVDF, PEO, and PSS, impeded the direct contact between ${\mathrm{PF}}_{6}^{-}$ in LE and the LFP. Additionally, the distributions of polymers and Li^+^ profiles exhibited apparent overlapping for the PEO and PSS systems, indicating a stronger polymer-Li^+^ interaction compared to the PVDF system. This suggests that PEO and PSS enhance the Li^+^ affinity toward LFP, while PVDF exhibits a less pronounced influence on Li^+^ distribution near LFP’s surface.

At the Li^+^-depleted FP surface, the concentration profiles revealed that polymers predominantly occupied the interface region in both the PVDF and PEO systems, effectively excluding both Li^+^ and the counter ions from the FP surface. This suggests that Li^+^ has weak affinity toward the FP surface, which cannot be attenuated by PEO binders. Interestingly, the PSS binder still attracted Li^+^ to distribute near the FP surface. This indicates that PSS can more effectively improve the Li^+^ distribution near the Li^+^-depleted FP interface than the PVDF and PEO binders.

Figure 3 displays the radial distribution function (RDF, g(r)) between the polymer and Li^+^ and the coordination number (C.N.) of Li^+^ within the interfacial region of the LE/cathode interface for all the tested binders. Here, the C.N. evaluated the average number of coordinating molecules within the cutoff distance $r_{cut}$ = 2.9 Å around Li^+^, corresponding to the average range of the first RDF peak. The C.N. was calculated from the RDF as:(1)$$\begin{array}{c} C.N.=\rho \int_{0}^{r_{cut}}4\pi r^{2}g\left(r\right)dr, \end{array}$$
where ρ denotes the bulk density of the coordinating atoms. From Figure 3a, we observed strong associations of Li^+^ with both PEO and PSS near the Li^+^-rich LFP surface, in contrast to a notably weaker PVDF-Li^+^ interaction. This indicates the significantly stronger interactions of PEO and PSS with Li^+^ compared to PVDF, consistent with the difference in the Li^+^ distributions near the LFP surface among the three tested binders observed in Figure 2. At the LE/FP interface, the RDF between the polymer and Li^+^ showed a diminished Li^+^ association of PEO, mainly due to the weak Li^+^ affinity of the FP surface. In contrast, the negatively charged PSS binder retained a high RDF intensity, indicating PSS maintains a high Li^+^ association that can overcome the weak Li^+^ affinity of the FP surface.

According to the Li^+^ coordination number (C.N.) analyses, as depicted in Figure 3b, Li^+^ primarily associated with EC and DEC of the electrolyte, particularly EC, within interfacial regions of both the LFP and FP surfaces for the PVDF system, where the polymer has minimal contribution within the Li^+^ coordination shell. Conversely, in the PEO and PSS systems, the coordinations of Li^+^ with EC and DEC were reduced, implying that the Li^+^ coordination with the LE solvent is partially disrupted by the PEO or PSS binders. Such an effect was more pronounced in the PSS system, where the total C.N. was reduced in the PSS system. This indicates that PSS can effectively disrupt the coordination shell of Li^+^, whereas PEO replaces the solvent coordinations within the Li^+^ coordination shell while retaining the total C.N. The results suggest that PSS has stronger effects on the electrochemical environment surrounding the Li^+^.

Figure 4 illustrates the mean squared displacement of Li^+^ within the interfacial regions at the LE/cathode interfaces for all tested polymer binders. The listed diffusion coefficient of Li^+^, $D_{{\mathrm{Li}}^{+}}$, in Figure 4 was evaluated based on the Einstein relation [40]:(2)$$\begin{array}{c} \lim_{t\to \infty }\langle \Delta r{\left(t\right)}^{2}\rangle =\lim_{t\to \infty }\langle {\left(r\left(t_{0}+t\right)-r\left(t_{0}\right)\right)}^{2}\rangle =6D_{Li^{+}}t, \end{array}$$
where $r(t_{0})$ denotes the position of the Li^+^ at time $t_{0}$, Δr(t) is the Li^+^ displacement after the passing time *t*, and the angle brackets represent the ensemble average. Note that the MSD analyzed the average $D_{{\mathrm{Li}}^{+}}$ for all Li^+^ in the interfacial region. Additionally, the crystalline Li^+^ at the LFP surface were excluded from the diffusivity calculation, allowing us to focus on the mobility of solvated Li^+^ near the cathode surface.

From Figure 4a, the $D_{{\mathrm{Li}}^{+}}$ in the PVDF system was greater than that in the PEO and PSS systems. The strong interaction of Li^+^ with PEO and PSS leads to the collective motion of Li^+^ with polymers and thereby reduces the average mobility of Li^+^. In contrast, Li^+^ has minimal interaction with PVDF and its mobility is therefore much more correlated to the LE solvent, i.e., EC and DEC. For the Li^+^-depleted FP surface, the $D_{{\mathrm{Li}}^{+}}$ in the PSS system is significantly smaller than that in the PVDF and PEO systems. As shown in Figure 2, Li^+^ remained strongly associated with PSS at the FP surface, whereas Li^+^ was nearly depleted within 7 Å of the FP surface for both PVDF and PEO. Additionally, less bound Li^+^ with higher mobility were observed in the range of 7 to 12 Å of the FP surface. This results also suggest a large variance in $D_{{\mathrm{Li}}^{+}}$ with respect to the position of Li^+^, mainly originating from the difference in the local environment at the LE/cathode interface.

### 3.2. Li^+^ Distribution and Diffusivity near Solid Polymer Electrolyte/Cathode Interface

To examine the effects of the host medium for electrolyte, we conducted MD simulations on solid polymer electrolyte (SPE)/cathode systems with the three tested polymer binders. Figure 5 illustrates the resulting concentration profiles of Li, polymers, and ${\mathrm{PF}}_{6}^{-}$ for the SPE/cathode systems with PVDF, PEO, and PSS binders. Note that the PEO system could be considered a binder-free system since the binders were essentially the polymer host of SPE. For the Li^+^-rich LFP surface, the anion TFSI^−^ exhibited low concentration near the LFP surface, similar to that observed in the LE/LFP systems. In contrast, the Li^+^ concentrations near the LFP surface for the PEO and PSS systems were much greater than that for the PVDF system, attributed to the high Li^+^ affinities of PEO and PSS. This also suggests that PEO and PSS can enhance the Li^+^ affinity toward LFP, while PVDF exhibits a less pronounced influence on Li^+^ distribution near the LFP surface. Compared with the LE/cathode systems, the anion was distributed more evenly in the SPE/cathode, indicating a reduced anion polarization at the SPE/cathode interface.

For the Li^+^-depleted FP surface, the PVDF binder and the PEO host of SPE predominately occupied the interfacial region, where no Li^+^ and TFSI^−^ were observed near the interface. The suggests that the low affinity toward the FP surface of Li^+^ is universal for both the LE and SPE systems. For the PEO systems, only TFSI^−^ was observed near the interface, whereas Li^+^ remained depleted near the FP surface. This indicates that PEO cannot attenuate the low FP affinity of Li^+^, similar to that observed in the LE systems. Interestingly, the PSS binder attracted Li^+^ close to the FP surface and repelled TFSI^+^, owing to its negative charges. This indicates that PSS can more effectively improve the Li^+^ distribution near the Li^+^-depleted FP interface than PVDF and PEO binders, which may benefit Li^+^ interfacial transport during the battery’s operation.

Figure 6 displays the radial distribution function (RDF) between the polymer and Li^+^, and the coordination number (C.N.) of Li^+^ in the interfacial region near the SPE/cathode interface for all the tested binders. Similar to the LE systems, the C.N. was evaluated within 2.9 Å of Li^+^, corresponding to the average range of the first RDF peak. From Figure 6a, PEO and PSS exhibit strong association with Li^+^ at both the LFP and FP surfaces. In contrast, PVDF exhibits a much weaker association with Li^+^, particularly near the FP surface where no Li^+^ was distributed in the interfacial region with no RDF peak observed. Compared with the SPE/LFP interface, the RDF peaks near the FP surface were significantly lower for the PEO system. This is due to the weak Li^+^ affinity to the FP surface, resulting in low Li^+^ concentration in the interfacial region as shown in Figure 5.

As shown in Figure 6b, the anion exhibited minimal contributions to Li^+^ C.N. near both LFP and FP surfaces, suggesting that Li^+^ was fully dissociated from the counter ion at the SPE/cathode interface. Additionally, the C.N. between Li^+^ and the PEO host of SPE was not affected by PVDF, and the total C.N. only sightly increased due to the weak association between Li^+^ and PVDF. This indicates that PVDF does not alter the coordination environment around Li^+^ with minimal changes on its chemical potential. In contrast, at both the SPE/LFP and SPE/FP interfaces, PSS reduced the C.N. between Li^+^ and PEO, while it only exhibited a C.N. of around 0.5 with Li^+^, leading to a significant decrease in the total C.N. This indicates that the electrostatic interaction between PSS and Li^+^ can effectively disrupt the coordination between Li^+^ and the PEO host of SPE, leading to changes in the electrochemical environment around Li^+^.

Figure 7 displays the mean squared displacement profiles of Li^+^ within the interfacial regions for all tested polymer binders on Li^+^-rich LFP or Li^+^-depleted surfaces. Among all three tested binders, the PEO system exhibited the highest $D_{{\mathrm{Li}}^{+}}$ near both the LFP and the FP surfaces, attributed to the polymer motions in the absence of incompatible polymer binders. Note that one of the major Li^+^ conduction mechanisms in PEO-based SPE is the hopping of Li^+^ among the coordination sites of PEO polymers [41,42,43]. Hence, Li^+^ can be conducted with the interference of polymers other than PEO, such as PVDF and PSS, leading to higher $D_{{\mathrm{Li}}^{+}}$. Conversely, within the PVDF system, Li^+^ mobility was significantly lower, owing to the inert Li^+^ affinity and the lack of the Li^+^ conducting ability of PVDF. For the PSS system, the resulting $D_{{\mathrm{Li}}^{+}}$ in the interfacial region was lower than that for the PEO system, due to the disruption of the coordination environment around Li^+^ affecting the Li^+^ hopping within SPE. Compared with the LE systems, the $D_{{\mathrm{Li}}^{+}}$ values near both the LFP and FP surfaces in SPE systems exhibited similar magnitudes, implying a more uniformed Li^+^ mobility during LIB operations.

### 3.3. The Effects of Binders on Li^+^ Local Diffusivity

The presented MSD analyses shown in Figure 4 and Figure 7 characterize the average Li^+^ mobility within the interfacial region near the cathode surface. Note, however, that binders do not completely cover the cathode surface, considering the low polymer loading during cathode fabrication. To better resolve the Li^+^ mobility on the cathode surface with non-uniform dispersion of the polymer and to characterize the local effects of binders, we analyzed the polymer lateral distribution and evaluated the $D_{{\mathrm{Li}}^{+}}$ for individual Li^+^ with respect to its position. Figure 8 illustrates the resulting lateral density profiles of polymers at the LE/cathode interfaces mapped with the Li^+^ positions in the interfacial regions emphasized by the corresponding $D_{{\mathrm{Li}}^{+}}$ values. For the PVDF system, the Li^+^ distribution was not correlated with the polymer coverage on both the LFP and FP surfaces. The Li^+^ diffusion occurred primarily in the non-polymer-covered region, due to the non-Li^+^-conducting nature of PVDF. Hence, a higher PVDF loading can lead to more polymer coverage on the cathode surface, hindering the Li^+^ interfacial transport. In contrast, the Li^+^ was distributed in the polymer-covered regions for the PEO system, owing to the strong coordination between Li^+^ and PEO. Hence, the Li^+^ diffusion was highly associated with the PEO on the cathode surface, where high Li^+^ diffusivity occurred in the PEO-covered regions. For the PSS system, the Li^+^ diffusivity was reduced compared with the PVDF and PEO systems. The most diffusive Li^+^ were located at the boundary of the PSS-covered region, indicating that the Li^+^ attracted via electrostatic interaction could move along the polymer surface. Notably, the Li^+^ remained highly mobile in the non-PSS-covered region. Compared with the Li-coordinating PEO, the negatively charged PSS exhibit a long-range electrostatic effect by casting an electric field on the cathode surface [21]. This allows PSS to affect Li^+^ mobility in the non-polymer-covered region, whereas PEO only affects the coordinated Li^+^ locally.

Figure 9 shows the lateral density profiles of polymers at the cathode interfaces mapped with the Li^+^ local diffusivities in the interfacial regions for the SPE/cathode systems. For the PVDF binder, similar to the LE systems, Li^+^ distribution was not correlated with the PVDF coverage near the LFP surface, and the Li^+^ diffusion occurred primarily in the non-PVDF-covered regions, due to its non-Li^+^-conductive nature and the low Li^+^ affinity. For the FP interface with PVDF binder, due to the absence of Li^+^ distribution, no Li^+^ diffusion was observed. For the PEO system, due to the uniformity between the binder and the polymer host of SPE, the cathode surface was fully covered with PEO at both the LFP and FP interfaces. The Li^+^ diffusivity was also the largest among all three tested binder systems. Note that the regions with higher PEO concentrations were associated with higher Li^+^ diffusivity, primarily due to the higher hopping probability among PEO chains to facilitate Li^+^ transport. For the PSS system, Li^+^ were mostly distributed near the boundary of the PSS-covered region. The occurrence of Li^+^ diffusion occurred in both the PSS-covered and non-PSS-covered regions, similar to the LE system. Additionally, the Li^+^ diffusivity appeared to be higher in the non-PSS-covered region. This suggested that PSS affects the Li^+^ interfacial transport mainly through long-range electrostatic interaction, in contrast to the localized Li^+^ coordination of PEO. Compared with the LE systems, the $D_{{\mathrm{Li}}^{+}}$ variance was smaller in the SPE systems, indicating a uniformed Li^+^ conduction within SPE.

## 4. Conclusions

In summary, our molecular dynamics (MD) simulations provided a detailed examination of the molecular effects exerted by Li^+^-coordinating binders and negatively charged binders on Li+ transport at the electrolyte/cathode interface in comparison to the conventional PVDF binder. The concentration profiles of Li^+^, polymer binders, and anions unraveled distinctive behaviors near both Li^+^-rich LFP and Li^+^-depleted FP surfaces. In particular, the negatively charged PSS emerged as a standout performer to significantly enhance Li^+^ distribution near the $Li^{+}$-depleted FP surface, showcasing its potential to improve Li^+^ interfacial transport during battery operation. The radial distribution function (RDF) and coordination number (C.N.) analyses revealed the potent interactions of PEO and PSS with Li^+^, disrupting Li^+^ coordination with electrolyte solvents and influencing the electrochemical environment around Li^+^ in liquid electrolyte (LE), whereas PSS remained effective in the SPE system.

Moreover, our study investigated the molecular effects of polymers on the local Li^+^ mobility near the cathode surface, offering insights into the effects of the non-uniform dispersion of binders. The lateral density profiles of polymers mapped with local Li^+^ diffusivity revealed how binders with distinct Li^+^ affinities influenced Li^+^ diffusion in both LE and SPE systems. In particular, PSS exhibited a unique ability to attract Li^+^ along its covered regions, influencing Li^+^ mobility even in the non-PSS-covered regions. This emphasizes the pivotal role of negatively charged binders in shaping local Li^+^ transport dynamics. These findings collectively highlight the complex molecular interactions between binders and Li^+^, revealing the importance of such interactions at the cathode/electrolyte interface. The comprehensive insights gained from this study contribute valuable knowledge for tailoring binder materials to enhance their performance, benefiting the design and optimization of LIBs.

## Figures and Tables

**Figure 1 polymers-16-00319-f001:**
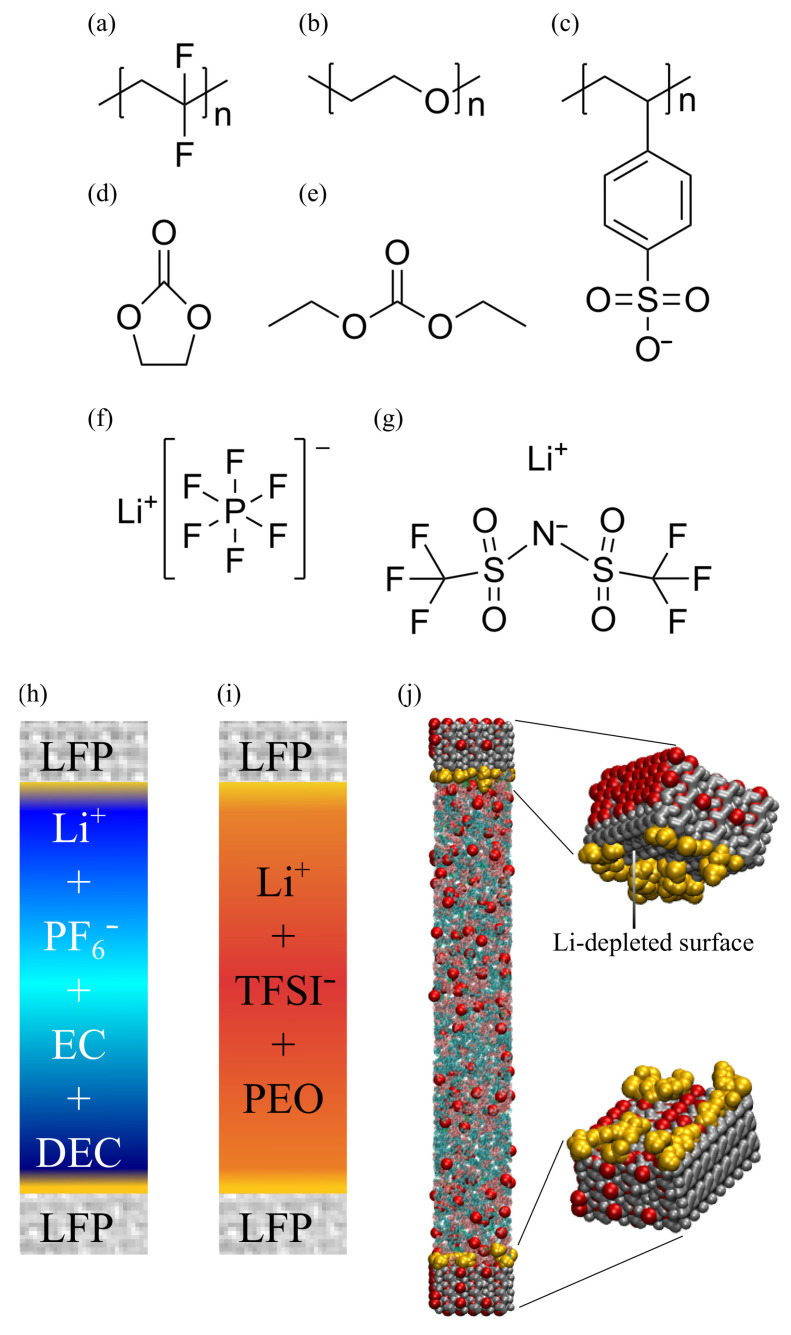
(**a**–**g**) Chemical structures for all polymers, molecules, and salts used in this work, including polyvinylidene fluoride (PVDF), polyethylene oxide (PEO), polystyrene sulfonate (PSS), ethylene carbonate (EC), diethyl carbonate (DEC), lithium hexafluorophosphate (LiPF_6_), and lithium bis(trifluoromethanesulfonyl)imide (LiTFSI). (**h**–**j**) Illustrations of an electrolyte/cathode system setup for an MD simulation. The cathode slab with electrolyte filled on its top and bottom, under the periodic boundary condition (PBC), is equivalent to two electrolyte/cathode interfaces at the top and the bottom of the simulation box. In this work, the bottom cathode surface was the Li^+^-rich LFP surface; the top cathode surface was the Li^+^-depleted FP surface.

**Figure 2 polymers-16-00319-f002:**
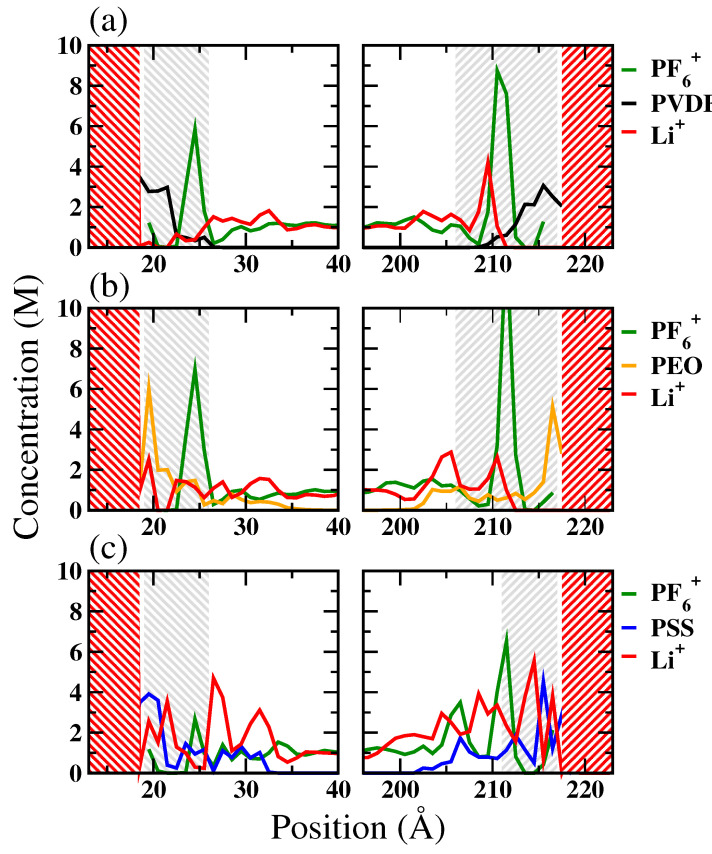
The concentration profiles along the interface normal, i.e., z-dimension, of Li^+^, polymer binders, and anion (${\mathrm{PF}}_{6}^{-}$) for the LE/cathode systems with (**a**) PVDF, (**b**) PEO, and (**c**) PSS binders. The red-shaded regions denote the cathode, and the gray-shaded regions denote the interfacial regions for further structural and dynamic analyses.

**Figure 3 polymers-16-00319-f003:**
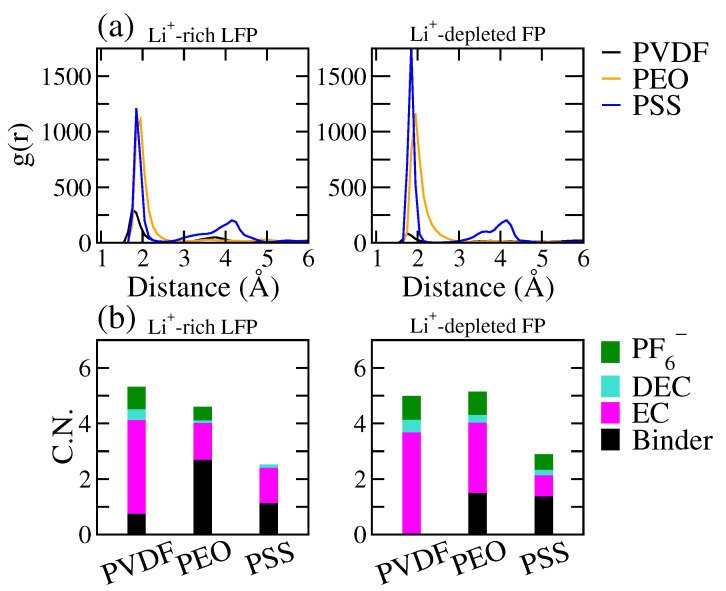
(**a**) The radial distribution function (RDF) between polymer and Li^+^ and (**b**) the coordination number (C.N.) of Li^+^ with surrounding molecules in the interfacial regions near the Li^+^-rich LFP (**right**) and Li^+^-depleted FP (**left**) surfaces of the LE/cathode systems with PVDF, PEO, and PSS binders.

**Figure 4 polymers-16-00319-f004:**
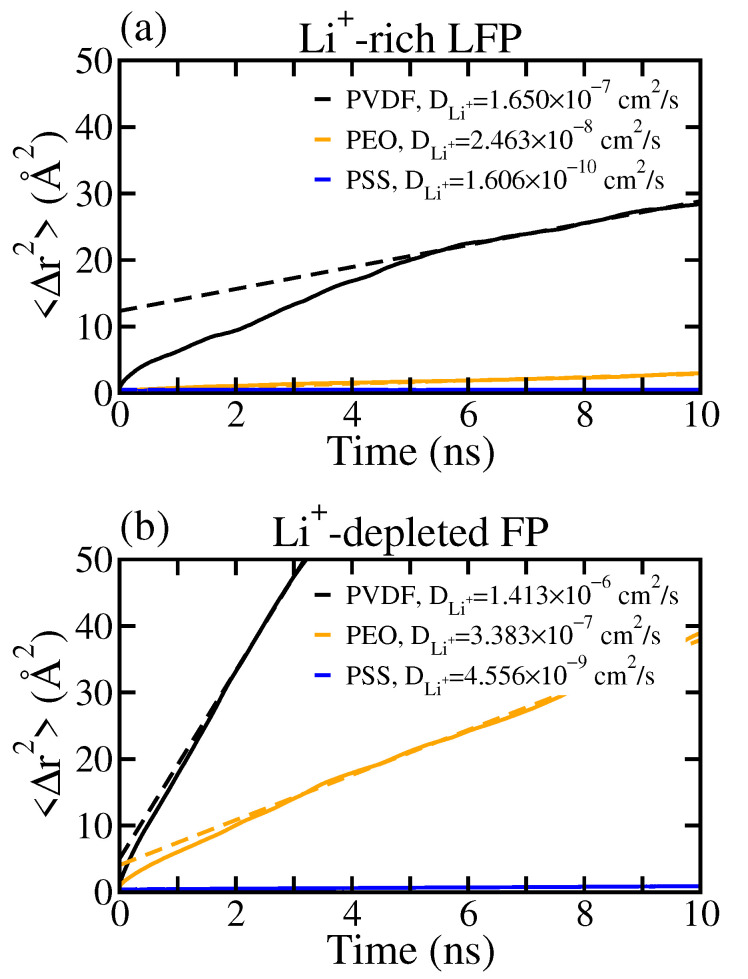
Mean square displacement $\langle \Delta r^{2}\rangle$ of Li^+^ within the interfacial regions for the LE/cathode systems with PVDF, PEO, and PSS binders near (**a**) the Li^+^-rich LFP surface and (**b**) the Li^+^-depleted FP surface. The dotted lines are the fitted straight lines for the diffusion coefficient evaluations.

**Figure 5 polymers-16-00319-f005:**
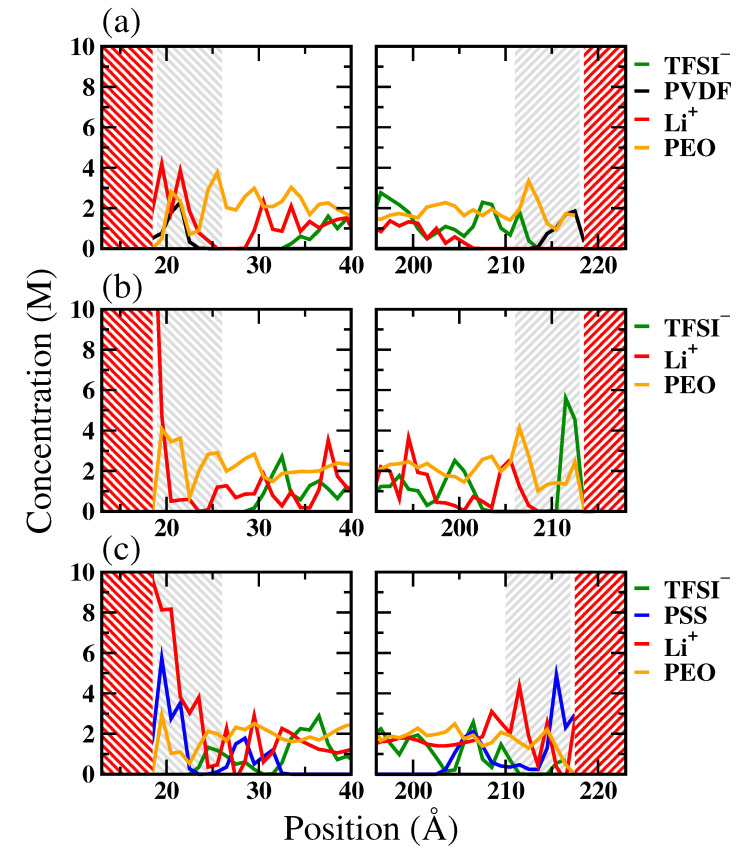
The concentration profiles along the interface normal, i.e., z-dimension, of Li^+^, polymer binders, and anion (TFSI^−^) for the SPE/cathode systems with (**a**) PVDF, (**b**) PEO, and (**c**) PSS binders. The red-shaded regions denote the cathode, and the gray-shaded regions denote the interfacial regions for further structural and dynamic analyses.

**Figure 6 polymers-16-00319-f006:**
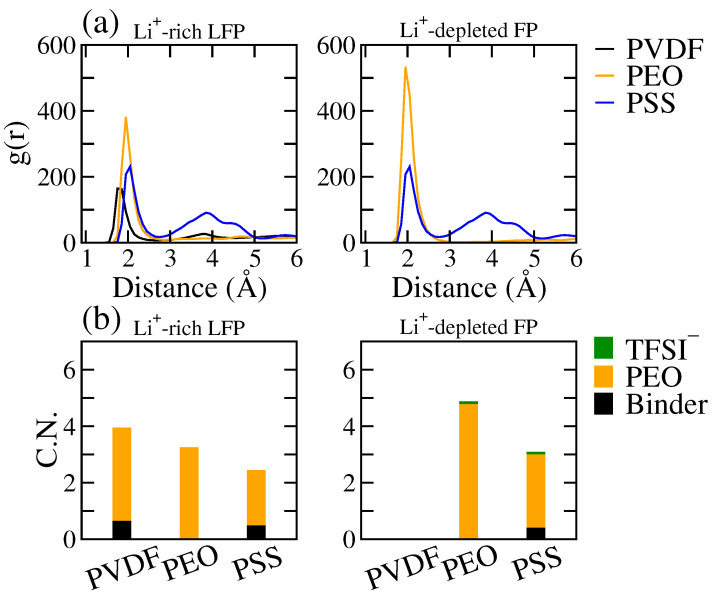
(**a**) The radial distribution function (RDF) between polymer and Li^+^ and (**b**) the coordination number (C.N.) of Li^+^ with surrounding molecules in the interfacial regions near the Li^+^-rich LFP (**right**) and Li^+^-depleted FP (**left**) surfaces of the SPE/cathode systems with PVDF, PEO, and PSS binders.

**Figure 7 polymers-16-00319-f007:**
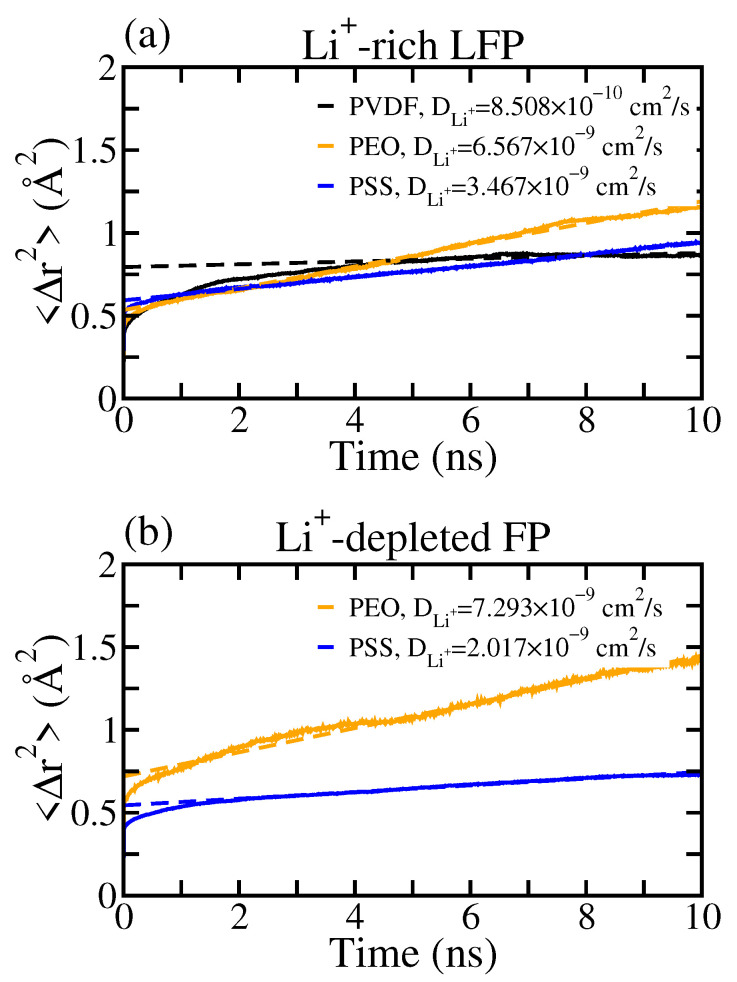
Mean square displacement $\langle \Delta r^{2}\rangle$ of Li^+^ within the interfacial regions for the SPE/cathode systems with PVDF, PEO, and PSS binders near (**a**) the Li^+^-rich LFP surface and (**b**) the Li^+^-depleted FP surface. The dotted lines are the fitted straight lines for the diffusion coefficient evaluations.

**Figure 8 polymers-16-00319-f008:**
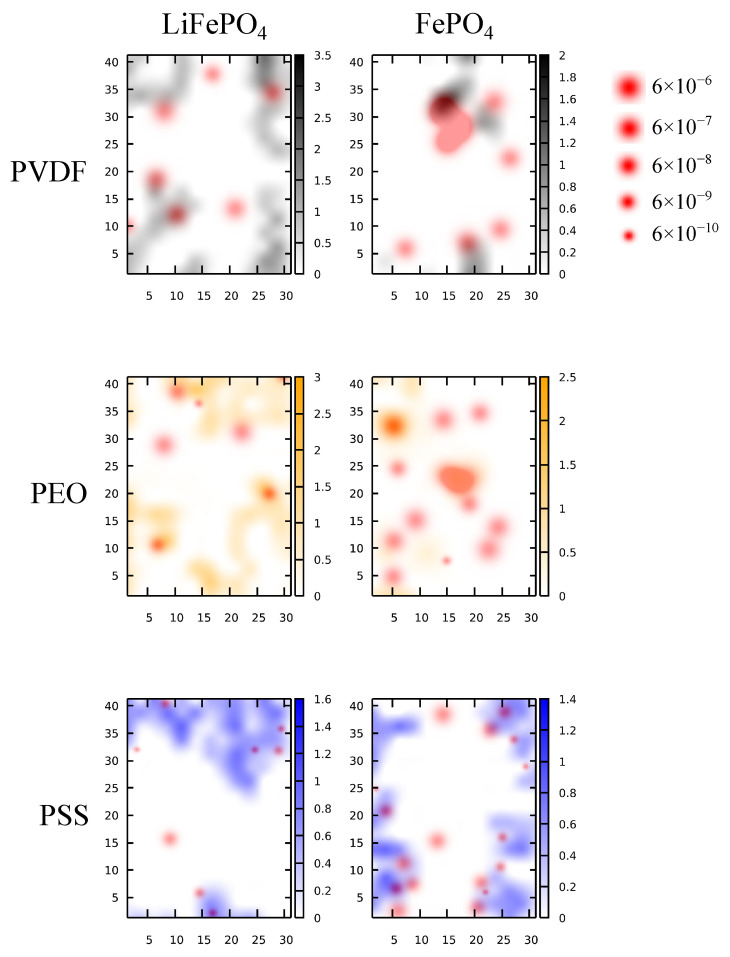
The polymer density heatmaps overlayed with the Li^+^ positions in the interfacial region near the Li^+^-rich LFP (**left**) and the Li^+^-depleted FP (**right**) surfaces, respectively, for the LE/cathode systems with PVDF (**top**), PEO (**middle**), and PSS (**bottom**) binders. The dot sizes for Li^+^ positions are scaled with the corresponding $D_{{\mathrm{Li}}^{+}}$ values.

**Figure 9 polymers-16-00319-f009:**
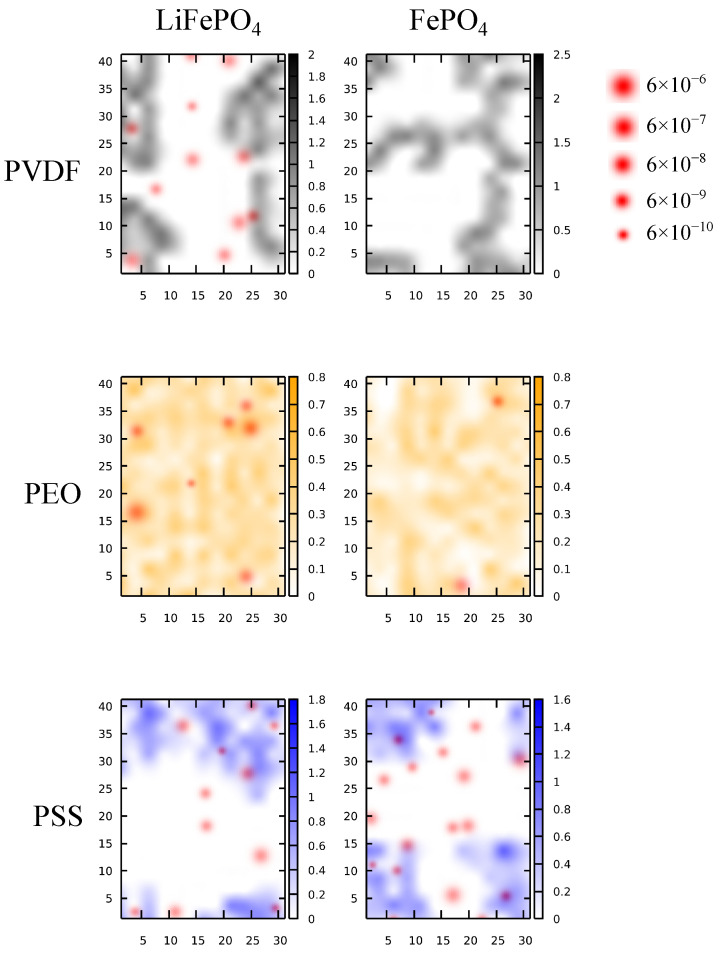
The polymer density heatmaps overlayed with the Li^+^ positions in the interfacial region near the Li^+^-rich LFP (**left**) and the Li^+^-depleted FP (**right**) surfaces, respectively, for the LE/cathode systems with PVDF (**top**), PEO (**middle**), and PSS (**bottom**) binders. The dot sizes for Li^+^ positions are scaled with the corresponding $D_{{\mathrm{Li}}^{+}}$ values.

## Data Availability

The data that support the findings of this study are available from the corresponding author, C.-c.C., upon reasonable request.

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
