# Peer review of "Molecular Effects of Li^+^-Coordinating Binders and Negatively Charged Binders on the Li^+^ Local Mobility near the Electrolyte/LiFePO_4_ Cathode Interface within Lithium-Ion Batteries"

_polymers, 2024, doi:10.3390/polym16030319_

Round 1

Reviewer 1 Report

Comments and Suggestions for Authors

Dear Authors,

In order to improve the overall quality of the article, please address the following comments:

1.      Line 28 and abstract: misplaced bracket in Poly vinylidene difluoride

2.      Line 34: misplaced bracket in poly acrylic acid

3.      Line 41: misplace bracket in poly ethylene glycol

4.      Line 78, 87,88: same issue

5.      Authors state “The amounts of binders introduced were based on the common experimental formulation of 5 - 10 % weight percents”: Is difference in weight of different binders influence the results of different binders’ performance?

6.      Please provide the equations used in the simulations? Provide rationale for assumptions taken during the simulations.

Comments on the Quality of English Language

Placement of brackets and thorough removal of typos needed.

Reviewer 2 Report

Comments and Suggestions for Authors

This work presented molecular dynamics for the impact from different binders on Li+ local mobility near the cathode interface. The work covered both liquid and solid electrolyte system, and would be important for the binder selections in LIB field. The manuscript is well-organized and written. I would suggest accepting it after addressing the following comments.

1. NMP is a common solvent for cathode fabrication. The author may want to specify the disadvantage of using PVDF with NMP. Why the aqueous-based fabrication is important? Except binder, is cathode material compatible with aqueous-based fabrication?

2. How was the coordination number of Li+ with different molecules determined in Figure 3?

3. Would the author please explain why PEO, PPS were selected in this work? Can you find experimental results from literatures to support your simulation conclusions?

Round 2

Reviewer 1 Report

Comments and Suggestions for Authors

No additional comments

Comments on the Quality of English Language

No comments